# Thermal Traits of MNPs under High-Frequency Magnetic Fields: Disentangling the Effect of Size and Coating

**DOI:** 10.3390/nano11030797

**Published:** 2021-03-19

**Authors:** David Aurélio, Jiří Mikšátko, Miroslav Veverka, Magdalena Michlová, Martin Kalbáč, Jana Vejpravová

**Affiliations:** 1Department of Condensed Matter Physics, Faculty of Mathematics and Physics, Charles University, Ke Karlovu 5, 121 16 Prague 2, Czech Republic; miroslavveverkag@gmail.com; 2J. Heyrovsky Institute of Physical Chemistry of the Czech Academy of Sciences, v.v.i., Dolejškova 2155/3, 182 23 Prague 8, Czech Republic; jiri.miksatko@jh-inst.cas.cz (J.M.); magdalena.michlova@jh-inst.cas.cz (M.M.); martin.kalbac@jh-inst.cas.cz (M.K.)

**Keywords:** magnetic nanoparticles, specific power absorption, magnetic fluid hyperthermia, surface coating, squid magnetometry, effective magnetic anisotropy

## Abstract

We investigated the heating abilities of magnetic nanoparticles (MNPs) in a high-frequency magnetic field (MF) as a function of surface coating and size. The cobalt ferrite MNPs were obtained by a hydrothermal method in a water–oleic acid–ethanol system, yielding MNPs with mean diameter of about 5 nm, functionalized with the oleic acid. By applying another cycle of hydrothermal synthesis, we obtained MNPs with about one nm larger diameter. In the next step, the oleic acid was exchanged for 11-maleimidoundecanoic acid or 11-(furfurylureido)undecanoic acid. For the heating experiments, all samples were dispersed in the same solvent (dichloroethane) in the same concentration and the heating performance was studied in a broad interval of MF frequencies (346–782 kHz). The obtained results enabled us to disentangle the impact of the hydrodynamic, structural, and magnetic parameters on the overall heating capabilities. We also demonstrated that the specific power absorption does not show a monotonous trend within the series in the investigated interval of temperatures, pointing to temperature-dependent competition of the Brownian and Néel contributions in heat release.

## 1. Introduction

Over the last decades, a lot of effort has been invested in the design of smart magnetic nanoparticles (MNPs) for a plethora of applications ranging from sensors [1,2,3], drug delivery [4,5], and tissue imaging [6] to magnetic fluid hyperthermia (MFH) [7,8,9,10].

In MFH, the high-frequency magnetic field (MF) is used to induce the particle’s macrospin reversal via Brownian and/or Néel relaxation mechanisms [11]. Both mechanisms usually take place within the dispersions of hyperthermic MNPs, and thus, they both contribute to the frequency- and amplitude-dependent heat dissipation [12,13,14]. Last but not least, the practical biomedical applications require a high degree of biocompatibility, which thus points to spinel ferrite MNPs such as magnetite, manganese, zinc ferrite, or their core-shell variants, where the outer shell is formed of a biocompatible ferrite [13]. There has been many efforts to find a universal explanation for the “best heaters”, based largely on particle size [15,16]. However, the heating efficiency and performance of the MNPs is the result of a delicate balance between the high-frequency MF parameters, viscosity, and heat capacity of the liquid medium, as well as the chemical, structural, morphological, and magnetic properties of the used MNPs [11,17,18,19,20]. The latter criterion is particularly intricate, considering the various possible scenarios of spin disorder that can be found [21]. On top of that, the mesoscopic effects, such as the formation of chains or aggregation, which can be reversible or irreversible, have a large impact on the heating efficiency [17,22].

It has also been reported that the enhanced heating by magnetically well-oriented samples is due to the greater effective anisotropy energy density [17,19] along with the linear chain-like structures, which is the origin of the larger dynamic hysteresis loop area [23]. Moreover, the mesoscopic changes in the MNP’s architecture within the dispersions, are closely related to the concentration [24] and surface coating (functionalization) [25,26,27,28,29], giving rise to the competition of steric effects and electrostatic attraction/repulsion and thus influencing the stability of the dispersion.

In this study, we explored the effects of different surface functionalizations, particle size, and mean magnetic moment, on the hyperthermic performance of cobalt ferrite (${\mathrm{CoFe}}_{2}O_{4}$) MNPs dispersed in 1,2-dichloroethane at the same concentration of 4 mg/ml. We systematically studied the heating performance under AC MFs in the frequency and amplitude ranges of 346–782 kHz and 0.01–0.06 T, respectively. For the study, we synthesized the ${\mathrm{CoFe}}_{2}O_{4}$ MNPs with different surface coatings and sizes. The critical size for the superparamagnetic transition appears to be ≈5 nm at around room temperature as a consequence of the strong magnetocrystalline anisotropy, which is also preserved for small MNPs [30,31,32,33,34].

As we have seen, surface coating functionalization is of great interest, in particular for biocompatible applications, since they can avoid particle aggregation, reduce toxicity, allow for drug delivery, click chemistry, etc.  [18,27,35,36] Here, the initial ${\mathrm{CoFe}}_{2}O_{4}$ MNPs were obtained by hydrothermal synthesis in a water–oleic acid–alcohol system, which gives us oleic acid (OA)-coated MNPs. Posteriorly, the surface coating OA was exchanged for 11-maleimidoundecanoic acid (MA) and 11-(furfurylureido)undecanoic acid (FU). For the second part of the study, a second hydrothermal treatment was applied onto the initially synthesized cobalt ferrite MNPs (seeds), yielding ${\mathrm{CoFe}}_{2}O_{4}$ MNPs with larger diameters.

All samples were investigated by high-resolution transmission electron microscopy (HR TEM), X-ray diffraction (XRD), and superconducting quantum interference device (SQUID) magnetometry. The ligand exchange was confirmed using Fourier transform infrared spectroscopy (FTIR), and the amount of the organic content was evaluated using the differential thermal analysis (DTA). We put in context the key magnetic parameters of the ${\mathrm{CoFe}}_{2}O_{4}$ MNPs, such as effective anisotropy ($K_{\mathrm{eff}}$), blocking temperature ($T_{B}$), and mean magnetic moment per particle ($\mu_{m}$), and discuss the relationship between these parameters and the heating performance.

## 2. Experimental Methods

### 2.1. Synthesis of MNPs

In our work, we aimed to obtain a representative series of samples with different surface coating while maintaining the size and chemical composition of the core intact. We also addressed the role of the particle size and structural/magnetic disorder effects by studying MNPs obtained by growing an extra ${\mathrm{CoFe}}_{2}O_{4}$ shell on top of the initial ${\mathrm{CoFe}}_{2}O_{4}$ seeds, which can be considered as cores. (The strategy is sketched in the Figure 1c).

The original OA-coated ${\mathrm{CoFe}}_{2}O_{4}$ MNPs were synthesized by a previously reported procedure based on the decomposition of iron (III) nitrate and cobalt (II) nitrate in the presence of sodium oleate (180 ${}^{\circ }$C, 11 h) [27,37]. The as-obtained product was dispersed in n-hexane and purified using repetitive precipitation with ethanol, followed by a final centrifugation that provides an uniform fraction of ${\mathrm{CoFe}}_{2}O_{4}$ MNPs. For the heating experiments and ligand exchange procedures, the MNPs were dispersed in 1,2-dichloroethane.

Inspired by the work of Angotzi et al. [38], we repeated the hydrothermal procedure and used the as-prepared ${\mathrm{CoFe}}_{2}O_{4}$ MNPs as seeds for the synthesis of larger core-shell nanostructures. Thus, a certain portion (a 50% seed percentage was used) of freshly prepared OA-coated ${\mathrm{CoFe}}_{2}O_{4}$ MNPs, placed in 10 mL of n-hexane, was added under vigorous stirring to the initial mixture of iron and cobalt nitrates and sodium oleate in a teflon tube. The tube was then placed in an autoclave, and the mixture was treated under the same conditions as in the first stage (180 ${}^{\circ }$C, 11 h) and purified as described previously. Using this method, we thus obtained a new batch of MNPs, which was then dispersed in n-hexane and purified as described above.

The OA-coated MNPs are further abbreviated as CoFe1_OA and CoFe2_OA according to one- and two-step synthesis, respectively.

As a next step, a ligand exchange was performed. Similar to a previous work [27], we exchanged the OA for two different ligands with similar dimensions: the MA and FU. Briefly, appropriate amounts of the ligands (1:2 molar ratio equivalent with respect to the OA content) were added to the dispersions of ${\mathrm{CoFe}}_{2}O_{4}$ MNPs with a known concentration into 1,2-dichloroethane. The homogenized dispersions were allowed to react for 24 h at room temperature and subsequently purified by dialysis in 1,2-dichloroethane. An aliquot of each dispersion of known volume was taken, evaporated to dryness in vacuum, and weighed. We repeated this method three times for each dispersion and calculated the average value of the mass concentration of the MNPs. These samples will be referred to from this point forward as CoFe_MA and CoFe_FU.

### 2.2. Characterization of MNPs

A successful ligand exchange was confirmed by diffuse reflectance infrared Fourier transform spectroscopy (DRIFTS) (see Appendix A). The infrared spectra were recorded on dispersions of the powderized sample in KBr using a Nicolet 6700 FTIR spectrometer with Happ–Genzel apodization in the 400–4000 ${\mathrm{cm}}^{-1}$ spectral region.

Hydrodynamic diameter of the MNPs in 1,2-dichloroethane was obtained using the dynamic light scattering method (DLS, Zetasizer Nano Malvern). The physical size and morphology of the MNPs were determined with the help of high-resolution transmission electron microscopy (HR TEM). The HR TEM images were acquired on a JEOL JEM 2100Plus microscope with acceleration voltage 200 kV. The droplets of samples (7 μL) were deposited on carbon-coated copper grids (300 mesh) and left to dry spontaneously in air. The picture analysis of the images was carried out using ImageJ 1.52a. The phase composition and particle size (by means of the coherently diffracting domain) were determined with the help of powder X-ray diffraction (XRD). The data were captured on fine powders deposited on a “zero diffraction” silicon plate using Bruker Advanced D8 diffractometer and analyzed using the WinPLOTR/FULLPROF package. Instrumental broadening was determined using the ${\mathrm{LaB}}_{6}$ standard.

The organic content in every batch of the MNPs was determined using Thermogravimetric Analysis (TGA) MS Netzsch STA449 F1 Jupiter equipped with Netzsch QMS 403 C Aelos mass detector (oxygen atmosphere).

The magnetic properties were measured by a magnetic property measurement system superconducting quantum interference device (SQUID) magnetometer (MPMS7 XL, Quantum Design, Inc., San Diego, CA, USA). In the experiment, the temperature range used was from 2 K up to 350 K for the magnetic susceptibility measurements and a field sweep from −7.0 T to 7.0 T for the magnetic isotherm assessment. In a typical experiment, a few mg of dry powder sample is placed in a gelatine capsule. First, the zero field and field cooled (ZFC-FC) curves were measured, followed by the magnetization isotherms at 10 K and 300 K. The obtained data was then used for determination of the basic magnetic parameters. In addition, the magnetic size, $d_{\mathrm{mag}}$, of the MNPs was calculated using Equation (4), present in the Appendix A. We present the values in the inverse and normal spinel limit to reflect the mixed character of the ${\mathrm{CoFe}}_{2}O_{4}$. The mean magnetic moment $\mu_{m}$ (please, see Appendix A for technical details) was refined using a LogNormal distribution fit (on OriginPro 2018) to the unhysteretic magnetization isotherms at 300 K, obtained from the data processed by the MINORIM software [39].

### 2.3. Heating Properties of MNPs

The heating properties of the MNPs were investigated using a D5 series G3 driver from NanoScale Biomagnetics [40]. The measurements were performed on 0.5 mL of ferrofluid (dispersion of MNPs in 1,2-dichloroethane adjusted to the Fe concentration of 4 mg/mL). The liquid was poured into a vial, which in turn was placed within a coil system connected to the G3 driver under adiabatic conditions. The temperature increase over time was registered by the G3 driver via a thermal optical probe immersed directly in the ferrofluid. The specific power absorption (SPA) [41,42] was determined for different frequencies (346–782 kHz) and amplitudes (0.01–0.06 T) of the applied high-frequency MF, by means of Equation (3) shown in the  Appendix A.

## 3. Results and Discussion

### 3.1. Basic Structural and Magnetic Properties

All samples were first subjected to structural and morphological analysis by HR TEM, XRD and DLS; the results are presented in Table 1, and the typical HR TEM images of CoFe1_OA and CoFe2_OA are shown in Figure 1a,b. The relative disorder difference seen between the HR TEM images (a) and (b) is due to the seeds prepared by the one-step process and introduced into the two-step synthesis of MNPs. The particle size derived from the HR TEM analysis shows comparable particle sizes for the MNPs based on the original CoFe1_OA core, while the CoFe2_OA sample reveals a size increase of about 0.6 nm. The hydrodynamic sizes in 1,2-dichloroethane vary only slightly, in correspondence with the physical (TEM) sizes. The XRD analysis pointed out the very good crystalline quality of the MNPs, whereas the relative decrease of the XRD size in relation to the TEM size is within the range usually observed for the MNPs prepared by this procedure [21,37]. The organic content, obtained from TGA is about 20% for all samples, which makes the comparative study more robust as the surface coverage by the ligands is more or less identical for all batches of MNPs. The covalent coupling of the ligands on the surface of the MNPs was confirmed by DRIFTS analysis (see Appendix A).

Turning our attention towards the basic magnetic characterization, Figure 2 presents the results of the magnetization measurements (isotherms at 10 K and 300 K), whereas a summary of the magnetic parameters is given in Table 2 and Table 3.

Upon closer inspection of the hysteresis loops at 10 K, there is no relevant variation in the coercivity either due to the functionalization process (Figure 2a,b) or due to the double synthesis one (Figure 2c,d). Only a slight difference in the saturation magnetization can be observed, which is within the experimental error of determination of the sample mass and its organic content. At 300 K, the magnetization curves are unhysteretic, as expected for superparamagnetic behavior. Again, there is no significant change in the overall magnetic properties due to either procedure. A slight change can be identified in the magnetic moment distribution (insets of Figure 2b,d), which reveals some differences in the mean magnetic moment values, summarized in Table 3. It is worth mentioning that the mean magnetic moments for CoFe1_OA and CoFe2_OA are very close while the XRD sizes are consistently different, which points to an extra spin disorder in CoFe2_OA, possibly at the internal “core-shell” interface, which gives rise to an extra contribution to the effective magnetic anisotropy. This scenario is also supported by a simple blocking temperature comparison of the MNPs and their dependence on the size and effective anisotropy. We expect that, for MNPs with very similar physical size and prepared by the same method, the effective anisotropy is not strikingly different. We can thus estimate the $T_{B}$ for the CoFe2_OA by considering the effective anisotropy value for CoFe1_OA and an average magnetic size, both in the mixed spinel structure calculated from the values given in Table 3. We obtain about 20 K lower $T_{B}$ for CoFe2_OA compared to the experimental value, which corroborates the extra contribution in the effective anisotropy. We point out that the internal structure of MNPs should not be neglected when interpreting the magnetic properties of MNPs, as it also governs the heating response [21].

The most striking difference can be observed in the blocking temperature, $T_{B}$ (determined using the derivative difference method [43]). While the coercivity is not changed much, the increase in $T_{B}$ is, by definition, attributed to the size (magnetic) and effective anisotropy of the ensemble of MNPs (please see Equation (5) in the SI for the ensemble of non-interacting MNPs). In our samples, the particle sizes are very similar and the spatial demands of the ligands are comparable; therefore, additional effects influencing the magnetic parameters (dipolar inter-particle interactions, spin canting due to the MNP proximity, etc.) are supposed to be the very same [44]. Thus the $T_{B}$ variation is driven mostly by the particle size and effective anisotropy changes. Please note that the $K_{\mathrm{eff}}$ values were calculated using $T_{B}$ values after Equation (6) in the Appendix A.

### 3.2. Heating Properties

To reach the main goal of our study, we carried out a systematic investigation of the heating abilities of the MNPs dispersed at the same concentration in 1,2-dichloroethane. Figure 3 shows the heating and SPA curves for all samples captured at 782 kHz and 0.03 T. More results for different frequencies and amplitudes of the MF are given in the SI. No morphological changes were seen on the nanoparticles (NPs) after the several heating cycles performed at different frequencies and MF amplitudes.

The time evolution of the temperature is shown in panel (a) of Figure 3. The maximum temperature achieved is for the largest MNPs—CoFe2_OA, which reaches the largest value compared to all other samples based on the CoFe1_OA “core”, as expected. The data also revealed that the process of ligand exchange hinders the achieved temperature limit for the CoFe_MA and CoFe_FU samples. At this point, we may consider the steric demands of the MA, FU, and OA ligands, which influence the Brownian contribution to heat generation. The more sterically demanding ligands (MA and FU compared to OA) are expected to enhance the friction between MNPs and/or the fluid, which is important in the Brownian mechanism (please see SI for a brief summary on the heating mechanisms). This scenario is supported in our series of samples, as we already demonstrated that the magnetic properties of the MNPs based on the same “core” are not significantly changed after ligand exchange. Please note that considering the steric effects is relevant only for the samples based on the same ”core”.

The trends in the SPA are shown in panel (b) of Figure 3. The MNPs with a slightly larger physical size, CoFe2_OA, do not show the best initial performance, while the CoFe_FU sample provides the best heating ability in that regard. Please note that we also inspected the role of aggregation, which is known to suppress the heating performance, and we did not observe any serious aggregation during and after the heating experiments.

These results are in slight controversy with previous works [15,16,45], which claim that the particle size is the most important parameter determining the heating capabilities of MNPs if other parameters (concentration, magnetic phase, etc.) are constant and the mesoscopic effects (aggregation and chaining) can be neglected or are expected to be comparable. If this is the case, CoFe2_OA should have the best heating properties throughout, as the sample has the largest size. To break through this controversy, we further correlated the magnetic parameters of the MNPs with the heating performance.

A comparison of the key magnetic parameters (for details, please see Section 3.1) and the SPA values (at two different temperatures) is presented in Figure 4. Several trends can be derived from the plots.

First, the relative change in SPA for the two quite close temperatures (20 ${}^{\circ }$C and 30 ${}^{\circ }$C) is more pronounced for the CoFe_FU and CoFe_MA samples. Second, the sample with the largest physical and magnetic size, CoFe2_OA, has lower initial SPA compared to CoFe_FU and CoFe_MA. Third, the best correspondence within the series can be found for the effective anisotropy, while the blocking temperature (function of the magnetic size and effective anisotropy) seems to have a random correlation.

In this vein, one expects that an increase in $K_{\mathrm{eff}}$ also increases the heating performance of the MNPs (Néel heating mechanism, see SI). This is reflected by the greater initial SPA value at 20 ${}^{\circ }$C. However, the heat release decreases when the dispersion reaches 30 ${}^{\circ }$C, and the SPA drops much more significantly for the CoFe_FU and CoFe_MA samples than that of the CoFe1_OA and CoFe2_OA. All the results thus indicate that the coating has a very important role in heating via influencing the friction and thus the Brownian contribution. On the other hand, the magnetic parameters determine the initial behavior, but they have a complex but less important role on the long time scale of the heating process when the temperature of the dispersion is much further above $T_{B}$ of the MNPs.

## 4. Conclusions

We studied the role of particle size and surface coating on heating performance under high-frequency MF for a series of cobalt ferrite MNPs based on the same “core”. The samples have either a different surface functionalization (OA, MA, and FU) or an extra magnetic shell of thickness comparable to the length of the ligands. We conclude that a simple correlation of the heating performance to the particle size is not sufficient and that the relation between the basic magnetic parameters, suggested in some previous works (e.g., [21]) is not consistent.

We can clearly distinguish between the initial and long-term phases of the heating process. While on the onset of heating, the explanation by means of $K_{\mathrm{eff}}$, $d_{\mathrm{mag}}$, and $T_{B}$ seems to be relevant for MNPs with the same type of coating. The trends in the heat release at elevated temperatures on large time scales does not show a clear correlation with the particle size and the basic magnetic parameters. The increases in effective anisotropy and magnetic size contribute both to the Néel and Brownian mechanisms, while the steric effects have an impact on the Brownian term only. For a narrow temperature interval, the samples with sterically demanding ligands show higher SPA values, which points to temperature-dependent competition of the two heating mechanisms. This particular finding is essential for all possible applications (magnetic fluid hyperthermia, MF-assisted chemical synthesis, and catalysis) where control of the temperature rise matters and may even become critical for the particular process.

## Figures and Tables

**Figure 1 nanomaterials-11-00797-f001:**
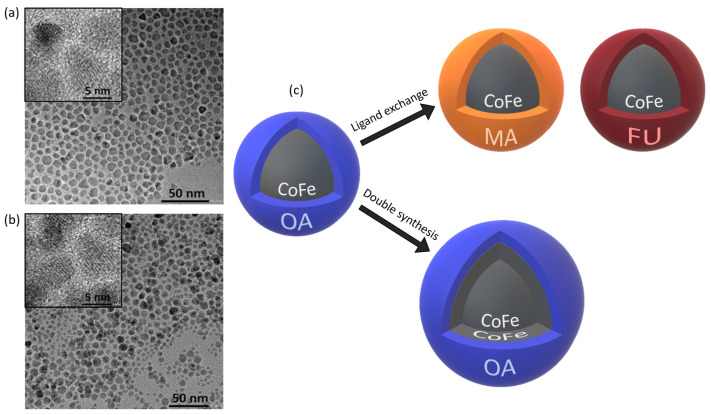
Representative high-resolution transmission electron microscopy (HR TEM) images of the obtained magnetic nanoparticles (MNPs) with insets showing their crystalline structure, for the one-step (**a**) and two-step (**b**) hydrothermal synthesis. Relative disorder difference is due to the 50% seed MNP usage in the two-step synthesis process. Panel (**c**) presents the synthesis and functionalization strategy for the MNPs used in the study. The initial MNPs covered with OA were either functionalized with MA or FU. As an alternative, the two-step process yields MNPs with larger size via synthesis on top of the initial cobalt ferrite core particles.

**Figure 2 nanomaterials-11-00797-f002:**
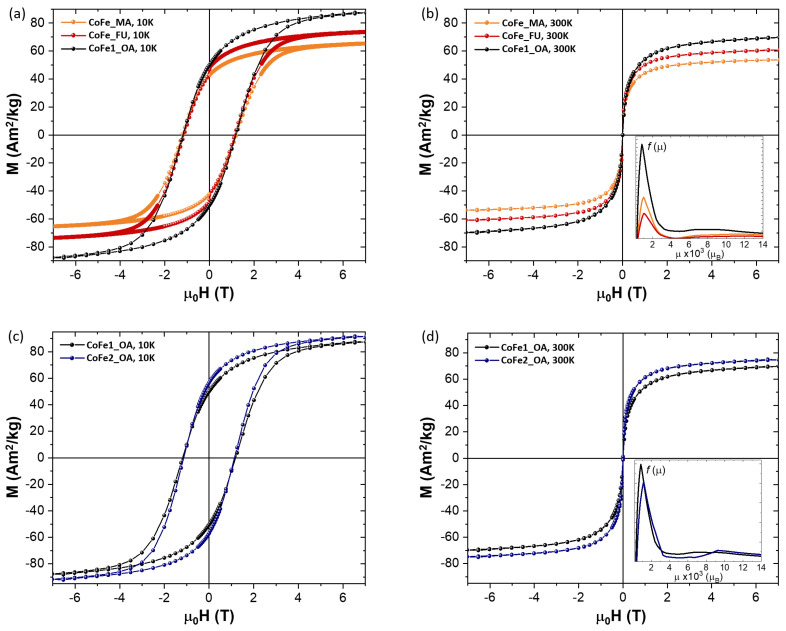
Magnetization isotherms measured at 10 K and 300 K. Panels (**a**,**b**) present the data for the MNPs based on one-step synthesis, while the panels (**c**,**d**) show the comparison between the MNPs obtained by the one-step and two-step protocols. The insets in panels (**b**,**d**) correspond to the magnetic moment distribution.

**Figure 3 nanomaterials-11-00797-f003:**
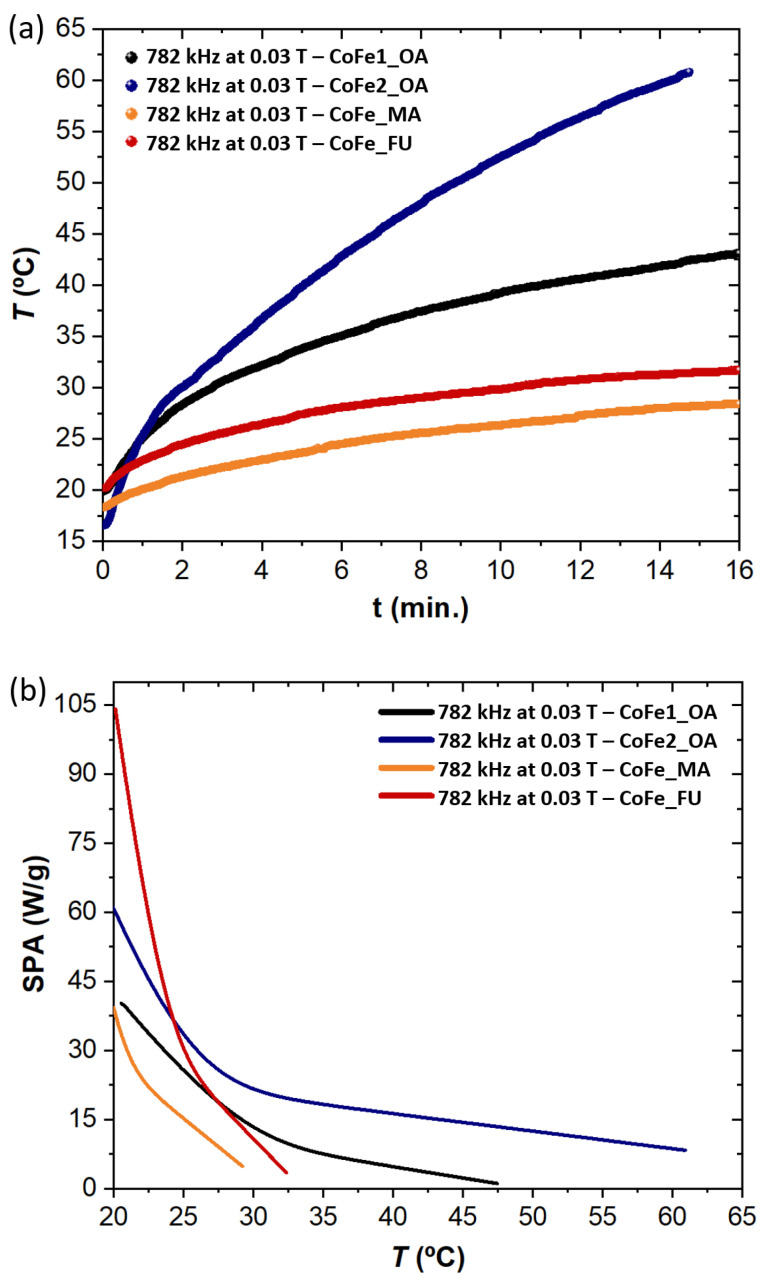
Comparison of the heating and specific power absorption (SPA) curves for the series of MNPs. (**a**) Temperature evolution over time for the different MNPs under the same parameters of high-frequency magnetic field (MF). (**b**) Evolution of SPA with the temperature. Please note that the results obtained for other tested frequencies/amplitudes can be viewed in the Appendix A.

**Figure 4 nanomaterials-11-00797-f004:**
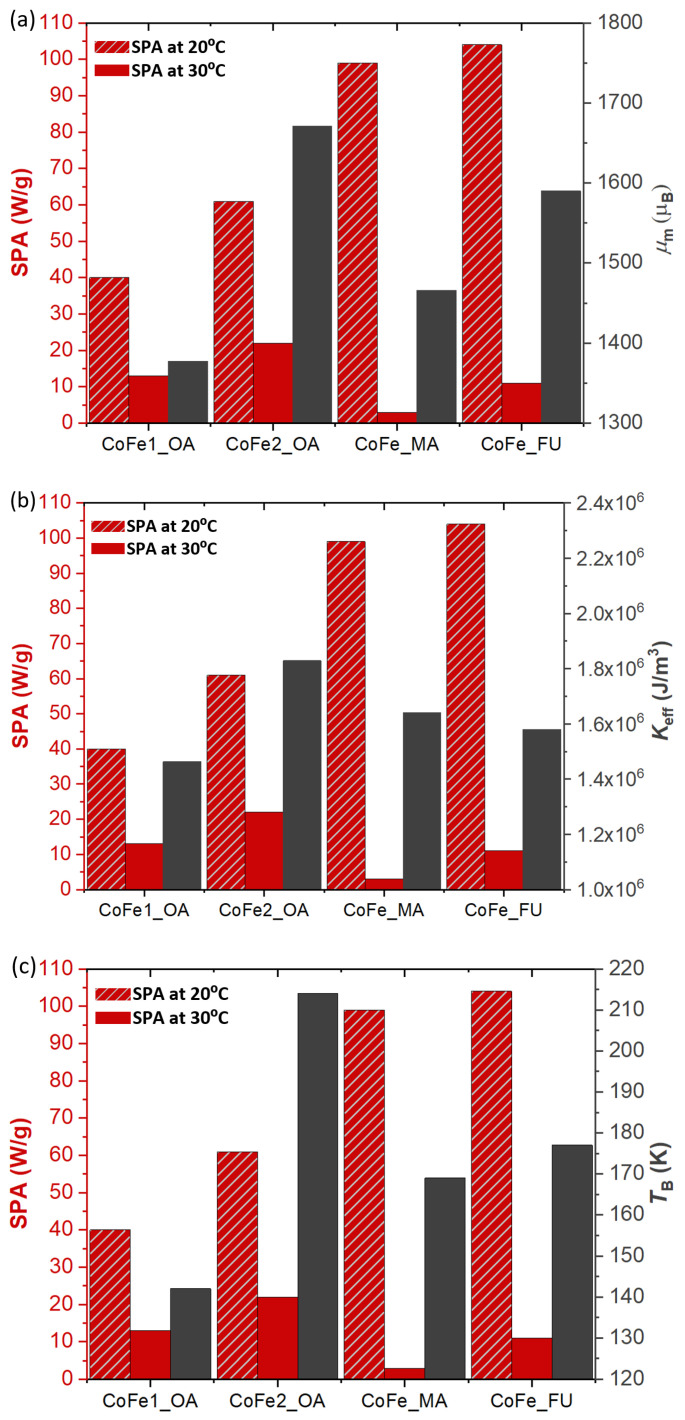
Correlation of the SPA with the characteristic magnetic parameters of the MNPs. (**a**) SPA at 20 ${}^{\circ }$C and 30 ${}^{\circ }$C compared to the mean magnetic moment, $\mu_{m}$. (**b**) SPA at 20 ${}^{\circ }$C and 30 ${}^{\circ }$C against the effective anisotropy, $K_{\mathrm{eff}}$. (**c**) SPA at 20 ${}^{\circ }$C and 30 ${}^{\circ }$C in comparison with the blocking temperature, $T_{B}$.

**Table 1 nanomaterials-11-00797-t001:** Basic properties of the MNPs: particle size (diameter in nm) obtained by TEM, dynamic light scattering method (DLS), and XRD together with the organic content (in percent) revealed by the Thermogravimetric Analysis (TGA).

	TEM (nm)	DLS (nm)	XRD (nm)	TGA (% of Coating)
CoFe1_OA	5.5±1.2	11.6±0.4	4.9±0.2	23±1
CoFe_MA	5.9±1.3	12.2±1.8	5.4±0.2	21±1
CoFe_FU	5.5±1.1	11.8±0.1	5.0±0.2	21±1
CoFe2_OA	6.1±1.5	13.9±1.5	5.6±0.3	20±1

**Table 2 nanomaterials-11-00797-t002:** Additional magnetic parameters derived from the hysteresis loops and zero field and field cooled (ZFC-FC) curves: $H_{c}$—coercive field at 10 K, $M_{r}$—remnant magnetization, $M_{S}$—saturation magnetization, and $T_{B}$—the blocking temperature. The $M_{r}$/$M_{S}$ ratio corresponds to the loop squareness.

	$H_{c}$ 10K (T)	$M_{S}$ 10K (${\mathrm{Am}}^{2}/\mathrm{kg}$)	$M_{S}$ 300 K (${\mathrm{Am}}^{2}/\mathrm{kg}$)	$M_{r}$/$M_{S}$ 10 K	$T_{B}$ (K)
CoFe1_OA	1.19	87	70	0.57	142
CoFe_MA	1.22	65	54	0.65	169
CoFe_FU	1.14	74	61	0.64	177
CoFe2_OA	1.14	91	75	0.62	214

**Table 3 nanomaterials-11-00797-t003:** Summary of the basic magnetic parameters of all samples, derived from the static magnetization measurements: $\mu_{m}$—mean magnetic moment per particle, $d_{\mathrm{mag}}$—magnetic size, and $K_{\mathrm{eff}}$—effective magnetic anisotropy, calculated for the inverse (Inv) and normal (Nor) spinel limits.

	$\mu_{m}\times 10^{3}$ ($\mu_{B}$)	$d_{\mathrm{mag}}^{\mathrm{Inv}.S}$ (nm)	$d_{\mathrm{mag}}^{\mathrm{Nor}.S}$ (nm)	$K_{\mathrm{eff}}^{\mathrm{Inv}.S}\times 10^{6}$ ($J/m^{3}$)	$K_{\mathrm{eff}}^{\mathrm{Nor}.S}\times 10^{6}$ ($J/m^{3}$)
CoFe1_OA	1.4±0.8	4.0	3.0	1.46	3.43
CoFe_MA	1.5±0.7	4.1	3.1	1.64	3.81
CoFe_FU	1.6±0.8	4.2	3.2	1.58	3.69
CoFe2_OA	1.7±0.9	4.3	3.2	1.83	4.27

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
