# Peer review of "Thermal Traits of MNPs under High-Frequency Magnetic Fields: Disentangling the Effect of Size and Coating"

_nanomaterials, 2021, doi:10.3390/nano11030797_

Round 1

Reviewer 1 Report

The authors present experimental studies on heating abilities of magnetic nanoparticles containing cobalt ferrites (CoFe2O4) as the magnetic core in dependence of different surface coatings and sizes. They use various methods and techniques to characterize the structural, chemical and magnetic properties as well as the performance of the nanoparticles concerning the specific power absorption. They show that the heating performance of the magnetic nanoparticles depends on various parameters and seems not be simply dependent on the size of magnetic nanoparticle as it is usually assumed. The drawback of the study is that the correlation between the different parameters as coating material and nanoparticle size do not show a comprehensive picture and it can be only argued with hand waving arguments about the interplay leading to the observed performance. Here it would have been interesting to investigate these magnetic particles with other techniques, e.g. neutron (and magnetic x-ray scattering) that could deliver more details about the magnetic structure and thus helps to shine a more profound light on potential spin arrangements that differ in these magnetic nanoparticles. Nevertheless, the variety of different techniques applied here leads a broad list of results that will deliver valuable input for future studies on the heat performance of magnetic nanoparticles. Thus, I recommend the paper for publications with some minor comments.

  1. Fig. 1: (a) and (b) show the TEM pictures for the one-step and two-step synthesis process and it seems that the details on (a) are better ordered as in (b). It would be good to comment on the differences, e.g. if this impression is right or it is owed to the selected area presented. If so, it should be explained in a little more detail how representative the TEM pictures are for (a) and (b).
  2. Fig. 2(b) and (d): The insets of the magnetic size contribution misses scale units (log, lin, etc.).
  3. Line 178: typo, please correct “crroborates” to “corroborates”

Author Response

We really appreciate the comments and relevance of the input. Unfortunately, we weren´t able to perform neither neutron nor magnetic x-ray scattering on the NPs (typically requires an acceptance of an experimental proposal a few months prior to scheduling the experiment). Moreover, the focus of the work was conceived on the SPA relation with the size, coating, anisotropy, magnetic moment, and blocking temperature, as determined from the SQUID isotherm measurements. Point by point reply to the comments is given below.

  1. Fig. 1: (a) and (b) show the TEM pictures for the one-step and two-step synthesis process and it seems that the details on (a) are better ordered as in (b). It would be good to comment on the differences, e.g. if this impression is right or it is owed to the selected area presented. If so, it should be explained in a little more detail how representative the TEM pictures are for (a) and (b)

The apparent higher order on (a) is due to that in (b), as stated in line 78, 50% of NPs are used as seeds for the two-step synthesis process, which increases the NP size. This means that the relative disorder is due to the presence of some NPs in the image (b) that are just single syntheses (smaller ones). Thus, both images are representative of the samples regarding their fabrication process. Moreover, we modified the image to include a zoomed-out inset of the HRTEM to show the overall NP distribution better.  Further clearing was added to the text, line 144, and the figure caption.

  1. Fig. 2(b) and (d): The insets of the magnetic size contribution misses scale units (log, lin, etc.).

These are graphs representing the normalized magnetic moment distribution and not magnetic size, and the scale is linear in Bohr magneton units on the x-axis (as shown). The y-axis represents just a number count of events that are usually omitted for better inset visualization.

  1. Line 178: typo, please correct “crroborates” to “corroborates”

Corrected.

Reviewer 2 Report

The manuscript presents on interesting study for different surface coating method in cobalt ferrite nanoparticle system. The relation between the particle size and magnetic performance under high-frequency MF was explored. Minor revision is required with the following comments.  

  1. The author should add more discussions for the previous studies in the introduction part.
  2. How did the author calculate the particle size?
  3. What's the particle morphology after the heating process?
  4. Please make sure use the same color to index the sample in the manuscript.

Author Response

We are grateful for the reviewer's comments, which we carefully considered in the revisions. Please, find the point-by-point reply below.

1. The author should add more discussions for the previous studies in the introduction part.

We extended the introduction and added several new references as suggested. The changes are tracked in the manuscript. 

2. How did the author calculate the particle size?

We are not sure about which particle size the reviewer is referring to. 

The NP sizes represented in Table 1 are based on TEM (statistical analysis of the TEM image using a ImageJ 1.52a software), DLS (calculated with the help of a routine embedded in the control software of the DLS Zetasizer Nano Malvern), and XRD experiments. For the XRD analysis, the Gaussian and Lorentzian components of the instrumental function were determined using the diffraction pattern of the LaB6 standard recorded for the identical instrumental set up, and the sample broadening was approximated by the Gaussian component only. In the structural model, the effects of spinel inversion approximated as a two-phase model were also tested. The magnetic size, d_mag, shown in Table 3, was calculated using equation (4) present on the Supplementary Information, SI, as stated in the main text section 2.2 Characterization of MNPs, line 123. The normal and inverse spinel limits were considered as the upper nad lower bound for the magnetic size, as they differ in the magnetic moment per unit cell, which is one of the parameters in equation (4).

3. What's the particle morphology after the heating process?

No change in the morphology of the NPs was observed by the DLS and TEM. A short note is added on line 199 of the main text.

4. Please make sure use the same color to index the sample in the manuscript.

Thank you for this point; we updated Figure 2 accordingly.

Reviewer 3 Report

The manuscript „ Thermal traits of MNPs under high-frequency magnetic fields: disentangling the effect of size and coating “ written by David Aurélio, Jiri Mikšátko, Miroslav Veverka, Magdalena Michlová, Martin Kalbác and Jana Vejpravová deals with heating abilities of magnetic cobalt ferrite nanoparticles in high-frequency magnetic fields. The particles prepared by hydrothermal synthesis with different surfactants in order to investigate the influence of hydrodynamic, structural and magnetic parameters on the heating capabilities. The manuscript is well structured and the results are clearly presented.

 Nevertheless, there are some aspects in which the manuscript did not convinced me in the present form. Here are my comments:

  1. Oleic acid is covalently bond to the particle surface, too. So, it is rarely possible to remove it from the particle. Please, consider the point of surfactant exchange more carefully. Why you don’t use a ligand which is weakly bond to the particle surface for the supplied particle “seeds”?
  2. How do you determine the concentration of particles in solution? In my opinion it is difficult to claim it exactly, because in the suspensions there is always an equilibrium state of dissolved metallic compounds and the in the particle cores crystallized ones. Furthermore, some of the original sample weight of metal salts is not transferred into the particle cores during synthesis and then washed away together with the redundant surfactants during the purification. This amount of Co/Fe cannot be neglected. Please, clarify this point.
  3. In Fig. 1 you showed bright field TEM images of your particles. This are nice overviews, but no high-resolution images. In several sections of your manuscript, you emphasize how important the microstructure and crystallinity of the particles is, regarding the heating capabilities, but the analysis of these feature is completely missing. Especially, for the “interface” between core and shell of the CoFe2_OA particles. Please, add these measurements.

Author Response

We are very grateful for the reviewer's comments. We addressed all of them in the revised version of our manuscript. A point-by-point reply is given below.

1. Oleic acid is covalently bond to the particle surface, too. So, it is rarely possible to remove it from the particle. Please, consider the point of surfactant exchange more carefully. Why you don’t use a ligand which is weakly bond to the particle surface for the supplied particle “seeds”?

There are quite a lot of reports on the ligand exchange on the surface of CoFe2O4 nanoparticles in the literature (to name a few: Chem. Mater. 2007, 19, 7, 1821–1831, Langmuir 2016, 32, 28, 7117–7126, J. Mater. Chem. C, 2020,8, 5417-5425, J Nanopart Res 19, 18 (2017), etc.). In our case, the type of coordination covalent bond Fe2+ COO- between CoFe2O4 particles and ligands is the same before and after ligand exchange. The driving force of the exchange is the excess of the new ligand and the whole process can take place in the sense of dynamic covalent chemistry. Its progress was evaluated by the DRIFTS spectroscopy. 

Following up on your comment, we have completed the analytical support presented by us in the SI. Therefore, we added three additional spectra demonstrating successful ligand exchange. Namely, Figure S3 - Comparison of DRIFTS spectra of CoFe_OA and CoFe functionalized with 11-maleimidoundecanoic acid (CoFe_MA), Figure S4 - DRIFTS spectrum of cobalt ferrites functionalized with 11-(furfurylureido)undecanoic acid and Figure S5 - Comparison of DRIFTS spectra of CoFe_OA and CoFe functionalized with 11-(furfurylureido)undecanoic acid (CoFe_FU). The spectra shows significant difference between CoFe particles originally coated with oleic acid and CoFe particles functionalized with 11-maleimidoundecanoic acid and 11-(furfurylureido)undecanoic acid, respectively.

2. How do you determine the concentration of particles in solution? In my opinion it is difficult to claim it exactly, because in the suspensions there is always an equilibrium state of dissolved metallic compounds and the in the particle cores crystallized ones. Furthermore, some of the original sample weight of metal salts is not transferred into the particle cores during synthesis and then washed away together with the redundant surfactants during the purification. This amount of Co/Fe cannot be neglected. Please, clarify this point.

We admit that an exact determination of the particle concentration is not an easy task. One theoretically possible approach is to measure UV/Vis spectroscopy, but dispersions with such a high concentration suitable for our heating experiments were too dark to obtain some reasonable results. Multiple dilutions would then introduce errors into the measurement. We aimed for a simple but reproducible method. Our approach was to take an aliquot of known volume, evaporate it to dryness in vacuum and weigh it using analytical balances. We repeated this method three times for each sample and calculated the average value of the mass concentration of particles. It should also be mentioned that after each ligand exchange, we thoroughly purified the dispersion by dialysis in a large excess of the solvent to remove both organic and inorganic residues.

We thank you for this comment, and therefore we added the following short explanation to the “Synthesis of MNPs” section (line 93) in the main text:

“...An aliquot of each dispersion of known volume was taken, evaporated to dryness in vacuum and weighed. We repeated this method three times for each dispersion and calculated the average value of the mass concentration of the MNPs.”

3. In Fig. 1 you showed bright field TEM images of your particles. This are nice overviews, but no high-resolution images. In several sections of your manuscript, you emphasize how important the microstructure and crystallinity of the particles is, regarding the heating capabilities, but the analysis of these feature is completely missing. Especially, for the “interface” between core and shell of the CoFe2_OA particles. Please, add these measurements.

Thank you for this remark. Indeed, we performed high-resolution TEM imaging on our samples, which in all cases confirmed the crystalline structure of the nanoparticles. We have modified Figure 1 accordingly. 

Round 2

Reviewer 3 Report

The manuscript has been clearly improved by the authors and is now ready for publication.